∂ | **Open Peer Review** | Antimicrobial Chemotherapy | Research Article

# Milteforan, a promising veterinary commercial product against feline sporotrichosis

Laura C. García Carnero,[1] Camila Figueiredo Pinzan,[1] Camila Diehl,[1] Patricia Alves de Castro,[1] Lais Pontes,[1] Anderson Messias Rodrigues,[2,3] Thaila F. dos Reis,[1] Gustavo H. Goldman[1,3]

**ABSTRACT** Sporotrichosis, the cutaneous mycosis most commonly reported in Latin America, is caused by the *Sporothrix* clinical clade species, including *Sporothrix brasiliensis* and *Sporothrix schenckii sensu stricto*. Due to its zoonotic transmission in Brazil, *S. brasiliensis* represents a significant health threat to humans and domestic animals. Itraconazole, terbinafine, and amphotericin B are the most used antifungals for treating sporotrichosis. However, many strains of *S. brasiliensis* and *S. schenckii* have shown resistance to these agents, highlighting the importance of finding new therapeutic options. Here, we demonstrate that milteforan, a commercial veterinary product against dog leishmaniasis, whose active principle is miltefosine, is a possible therapeutic alternative for the treatment of sporotrichosis, as observed by its fungicidal activity *in vitro* against different strains of *S. brasiliensis* and *S. schenckii*. Fluorescent miltefosine localizes to the *Sporothrix* cell membrane and mitochondria and causes cell death through increased permeabilization. Milteforan decreases *S. brasiliensis* fungal burden in A549 pulmonary cells and bone marrow-derived macrophages and also has an immunomodulatory effect by decreasing TNF-α, IL-6, and IL-10 production. Our results suggest milteforan as a possible alternative to treat feline sporotrichosis.

**IMPORTANCE** Sporotrichosis is an endemic disease in Latin America caused by different species of Sporothrix. This fungus can infect domestic animals, mainly cats and eventually dogs, as well as humans. Few drugs are available to treat this disease, such as itraconazole, terbinafine, and amphotericin B, but resistance to these agents has risen in the last few years. Alternative new therapeutic options to treat sporotrichosis are essential. Here, we propose milteforan, a commercial veterinary product against dog leishmaniasis, whose active principle is miltefosine, as a possible therapeutic alternative for treating sporotrichosis. Milteforan decreases *S. brasiliensis* fungal burden in human and mouse cells and has an immunomodulatory effect by decreasing several cytokine production.

**KEYWORDS** *Sporothrix brasiliensis*, *Sporothrix schenckii*, sporotrichosis, milteforan, miltefosine, antifungal agent, drug repurposing

Address correspondence to Gustavo H. Goldman, ggoldman@usp.br.

The authors declare no conflict of interest.

See the funding table on p. 15.

Sporotrichosis, a chronic cutaneous and subcutaneous infection, is the most commonly reported mycosis in Latin America and Asia, with a high prevalence in tropical and subtropical areas, including Brazil, Mexico, Argentina, India, Japan, and China (1, 2). Since 1998, Brazil has experienced large outbreaks of sporotrichosis that have been expanding throughout the country, mainly in the southeastern regions, the reason for which Brazil is considered a hyperendemic area (3–5).

Until 2007, *Sporothrix schenckii* was assumed to be the unique etiological agent for sporotrichosis, but recent molecular analyses have revealed the existence of several sibling species capable of causing infection (6). These species comprise the *S.*

*schenckii* clinical/pathogenic clade, which includes *S. schenckii sensu stricto*, *S. brasiliensis*, *Sporothrix globosa*, and *Sporothrix lurei* (7, 8). These species are thermodimorphic fungi, with a mycelial phase that grows in decaying organic matter at 25°C and a yeast phase that develops inside the host during infection (9, 10). The virulence profile varies among the species of the pathogenic clade being *S. brasiliensis* the most virulent, followed by *S. schenckii,* both with the capacity to cause severe infection even in immunocompetent individuals, whereas *S. globosa* and *S. luriei* are classified as low virulent species (11, 12).

Sporotrichosis can present different clinical manifestations, such as cutaneous (lymphocutaneous and fixed cutaneous), disseminated cutaneous, and extracutaneous (pulmonary, osteoarticular, ocular, meningeal, and visceral) (13). The development of one or other clinical forms depends on different factors, which include the host immune competence, site and depth of inoculation, amount of inoculum, and the etiological agent, all of which should be considered for proper patient management (14).

The transmission of the *Sporothrix* species is through traumatic implantation with contaminated material, the sapronosis, which is the classical route. However, in hyperendemic zones, such as Brazil, zoonotic infection by *S. brasiliensis* is highly reported, transmitted mainly by cats through scratching, biting, and even through contact with fluids from infected animals. This zoonotic transmission is considered a severe health problem in Brazil, especially in the area of Rio de Janeiro, due to the rapid spread of *S. brasiliensis*, which is associated with severe clinical manifestations in both humans and cats (15–18). Besides cats, dogs, albeit to a lesser extent, have also been affected by sporotrichosis, making this infection a significant veterinarian problem. Five thousand one hundred thirteen cases of feline sporotrichosis (from 1988 to 2017) and 244 canine cases (from 1988 to 2014) have been reported by the Evandro Chagas National Institute of Infectious Diseases in Rio de Janeiro, Brazil. However, this number is likely underestimated because sporotrichosis incidence is a mandatory notification only in a few states of Brazil (18).

Identification of the sporotrichosis causative agent is essential for treatment since the *Sporothrix* species show different antifungal susceptibility profiles (19–21), but this is not always possible, given that the identification of the species requires molecular tools (8). In general, for the treatment of cutaneous forms, itraconazole (ITZ) is considered the gold standard, whereas amphotericin B (AMB) is the first-line antifungal therapy used for disseminated forms (22, 23). However, in the last few years, many *S. brasiliensis* clinical strains have shown resistance to both azoles and AMB (24–26), complicating sporotrichosis treatment.

Miltefosine (MFS), also known as hexadecyl phosphocholine, is a synthetic glycerol-free phospholipid analog initially used as an antineoplastic drug (27, 28). Nowadays, MFS is the only available oral drug used to treat visceral and cutaneous leishmaniasis in dogs and humans due to its significant antiparasitic activity, *in vitro* and *in vivo*, against *Leishmania* species (29–32). MFS's action mechanism(s) has yet to be entirely understood. However, it has been demonstrated to act as a multi-target drug associated with the disruption of many vital pathways, such as (i) the inhibition of the biosynthesis of phosphatidylcholine, which causes low levels of this phospholipid (33, 34); (ii) the interference of the cell membrane calcium channels, which induces an increase of intracellular $Ca^{2+}$ (35, 36); (iii) the inhibition of the sphingomyelin biosynthesis, which increases ceramide concentration (37), resulting in cell apoptosis; and (iv) the immune response, in which its immunomodulatory effects induce the activation of the Th1 response, mainly through the increased production of IFNγ and IL-12, which prevails over the Th2 response driven by *Leishmania sp* (38).

MFS has also been reported as an antifungal agent *in vitro* against some of the most clinically significant pathogenic and opportunistic fungi, such as *Candida* spp., *Aspergillus* spp., *Fusarium* spp., and *Cryptococcus* spp. (39–44). In addition, it was recently shown that MFS has *in vitro* fungicidal activity against *Sporothrix* spp., inhibiting the growth of the mycelial phase of *S. brasiliensis*, *S. schenckii*, and *Sporothrix globosa* (45), and the yeast phase of *S. brasiliensis* strains resistant to (ITZ) and AMB (46). It was also demonstrated

that alone or in combination with potassium iodide, MFS inhibits the biofilm formation of *S. brasiliensis*, *S. schenckii*, and *S. globosa* (47, 48). All of this evidence suggests the potential of MFS for treating sporotrichosis. Repurposing orphan drugs, which is the application of existing drugs for different therapeutic purposes than the ones initially marketed for, is a good alternative for treating infections caused by susceptible or resistant microorganisms (49). Such is the case of MFS, which, besides being repurposed for treating leishmaniasis, has been recently designated for treating primary amebic meningoencephalitis and invasive candidiasis (50).

Here, we demonstrate that MFS has fungicidal *in vitro* activity against both morphologies (hyphae and yeast) of different *S. brasiliensis* and *S. schenckii* strains. We also showed that milteforan (ML), a commercial veterinary product against dog leishmaniasis, whose active principle is miltefosine (Virbac), can inhibit and kill *Sporothrix* species *in vitro*. ML treatment also increases the killing of *S. brasiliensis* yeast by the epithelial cells A549 and bone marrow-derived macrophages (BMDMs). Our results suggest ML as a possible veterinary alternative to treat feline sporotrichosis.

## RESULTS

### ML and MFS have fungicidal activity against *Sporothrix* spp. *in vitro*

The *in vitro* antifungal activity of several drugs against six strains of *S. schenckii* and *S. brasiliensis*, three from each species, were assessed according to their MIC and MFC values for the mycelial and yeast phases (Table 1). From these drugs, ITZ has already been reported to show fungistatic activity against *Sporothrix* spp., whereas terbinafine (TRB), AMB, and MFS are fungicidal drugs (19, 23, 24). On the other hand, voriconazole (VCZ) was reported to show low activity in inhibiting *Sporothrix* growth, whereas caspofungin (CSP) does not exhibit antifungal activity *in vitro* (20).

Similar to previous reports, we found that none of the *Sporothrix* strains, in either yeast or mycelium states, were inhibited by CSP or VCZ. At the same time, both morphologies from all the isolates were sensitive to low concentrations of TRB and AMB (MIC ≤2 µg/mL). For ITZ, all strains' conidia were highly resistant (MFC >8 µg/mL). At the same time, the yeast phase was more sensitive with MIC and MFC values ≤ 2 µg/mL, except the *S. brasiliensis* clinical isolate 4823 yeast phase, which shows resistance to the drug (MFC >8 µg/mL), as already reported (51). TRB and AMB present fungicidal activity against *Sporothrix* species, whereas ITZ is a fungistatic drug (Table 1). MFS and ML also have fungicidal activity *in vitro* against both morphologies from the *S. schenckii* and *S. brasiliensis* strains, with MIC and MFC values ≤ 2 µg/mL (Table 1; Fig. 1).

Once we showed the antifungal activity of MFS and ML against *Sporothrix* spp., we evaluated their ability to interact with some of the drugs already being used for treating sporotrichosis. MIC and MFC values of CSP, VCZ, ITZ, TRB, and AMB in combination with half MIC of MFS or ML were determined for the yeast morphology of each *Sporothrix* strain (Table 2). No differences in the activity of CSP and VCZ were observed, since neither of these drugs could inhibit *S. schenckii* or *S. brasiliensis* growth in the presence of MFS or ML. On the other hand, the interaction of MFS or ML with either ITZ, TRB, or AMB increases the antifungal activity against all of the *Sporothrix* strains tested, decreasing their MIC and MFC values.

Next, in order to determine what kind of interaction MFS has with ITZ, TRB, and AMB, the drug combination responses were analyzed using checkerboard assays and SynergyFinder software (52), which evaluates the potential synergy of two or more drugs. The dose-response data obtained when combining MFS with either TRB, ITZ, or AMB against *S. brasiliensis* and *S. schenckii* yeast cells show a likely additive interaction (synergy score from −10 to 10) (Fig. 2). As previously reported for ITZ (46), we found that MFS does not synergize with the drug against *S. brasiliensis* and *S. schenckii*.

**TABLE 1** MIC and MFC values of several antifungals against *S. schenckii* and *S. brasiliensis* yeast and mycelial phases[a]

| | | | CSP (4–0.06 µg/mL) | VCZ (4–0.06 µg/mL) | ITZ (8–0.125 µg/mL) | MFS (16–0.25 µg/mL) | ML (16–0.25 µg/mL) | TRB (4–0.06 µg/mL) | AMB (8–0.125 µg/mL) |
|---|---|---|---|---|---|---|---|---|---|
| *Ss* 4820 | Y | MIC | >4 | >4 | 0.25 | 2 | 2 | 1 | 2 |
| | | MFC | >4 | >4 | 0.5 | 2 | 2 | 1 | 2 |
| | M | MIC | >4 | >4 | 2 | 2 | 2 | 1 | 2 |
| | | MFC | >4 | >4 | >8 | 2 | 2 | 1 | 2 |
| *Ss* 4821 | Y | MIC | >4 | >4 | 0.5 | 2 | 2 | 1 | 2 |
| | | MFC | >4 | >4 | 2 | 2 | 2 | 1 | 2 |
| | M | MIC | >4 | >4 | 1 | 2 | 2 | 1 | 2 |
| | | MFC | >4 | >4 | >8 | 2 | 2 | 1 | 2 |
| *Ss* 4822 | Y | MIC | >4 | >4 | 0.125 | 2 | 2 | 0.5 | 2 |
| | | MFC | >4 | >4 | 0.25 | 2 | 2 | 0.5 | 2 |
| | M | MIC | >4 | >4 | 2 | 2 | 2 | 1 | 2 |
| | | MFC | >4 | >4 | >8 | 2 | 2 | 1 | 2 |
| *Sb* 4823 | Y | MIC | >4 | >4 | 2 | 2 | 2 | 0.5 | 2 |
| | | MFC | >4 | >4 | >8 | 2 | 2 | 0.5 | 2 |
| | M | MIC | >4 | >4 | 1 | 2 | 2 | 1 | 2 |
| | | MFC | >4 | >4 | >8 | 2 | 2 | 1 | 2 |
| *Sb* 4824 | Y | MIC | >4 | >4 | 0.5 | 2 | 2 | 0.5 | >8 |
| | | MFC | >4 | >4 | 1 | 2 | 2 | 0.5 | >8 |
| | M | MIC | >4 | >4 | 2 | 2 | 2 | 1 | 2 |
| | | MFC | >4 | >4 | >8 | 2 | 2 | 1 | 2 |
| *Sb* 4858 | Y | MIC | >4 | >4 | 0.125 | 2 | 2 | 0.125 | 2 |
| | | MFC | >4 | >4 | 2 | 2 | 2 | 0.125 | 2 |
| | M | MIC | >4 | >4 | 1 | 2 | 2 | 1 | 2 |
| | | MFC | >4 | >4 | >8 | 2 | 2 | 1 | 2 |

[a]*Ss*: *S. schenckii*, *Sb*: *S. brasiliensis*, Y: yeast phase, M: mycelial phase; CSP: caspofungin, VCZ: voriconazole, ITZ: itraconazole, MFS: miltefosine, ML: milteforan, TRB: terbinafine, AMB: amphotericin B.

## MFS localizes to the *Sporothrix* cell membrane and mitochondria and causes cell death

Although the antifungal effect of MFS against *Sporothrix* has been reported, the localization of the drug in the yeast is still unknown. In *Leishmania* (53) and *A. fumigatus* (43), MFS localizes in the cell membrane and the mitochondria, increasing mitochondrial fragmentation and damage. Here, we found that in *S. brasiliensis*, fluorescent MFS is also localized in the cell membrane and the mitochondria in 47% of the cells investigated (three repetitions of 100 cells each), as shown by MitoTracker colocalization (Fig. 3).

Subsequently, to evaluate the viability of the yeast in the presence of MFS, drug-treated cells were stained with propidium iodide (PI) and analyzed by fluorescence microscopy. Since PI only penetrates cells with damaged membranes, PI+ cells are considered to be going through late apoptosis or early necrosis (54). Treatment of *S. brasiliensis* yeasts with 2, 4, and 8 µg/mL of MFS shows dose-dependent damage of the cells since the PI signal increased with the drug concentration, with a significant difference at 8 µg/mL (Fig. 4), as early as 6 h of exposure, confirming the MFS fungicidal activity against *Sporothrix*.

## ML decreases *S. brasiliensis* fungal burden in A549 pulmonary cells and bone marrow-derived macrophages (BMDM)

To determine the antifungal activity of ML against *S. brasiliensis* in the host tissues, two cell lines were used: lung A549 cells and bone marrow-derived macrophages (BMDMs). As shown in Fig. 5a, ML concentrations from 5 to 40 µg/mL did not reduce A549 cell viability compared with the control. A549 cells were challenged with 1:10 and 1:20 ratios

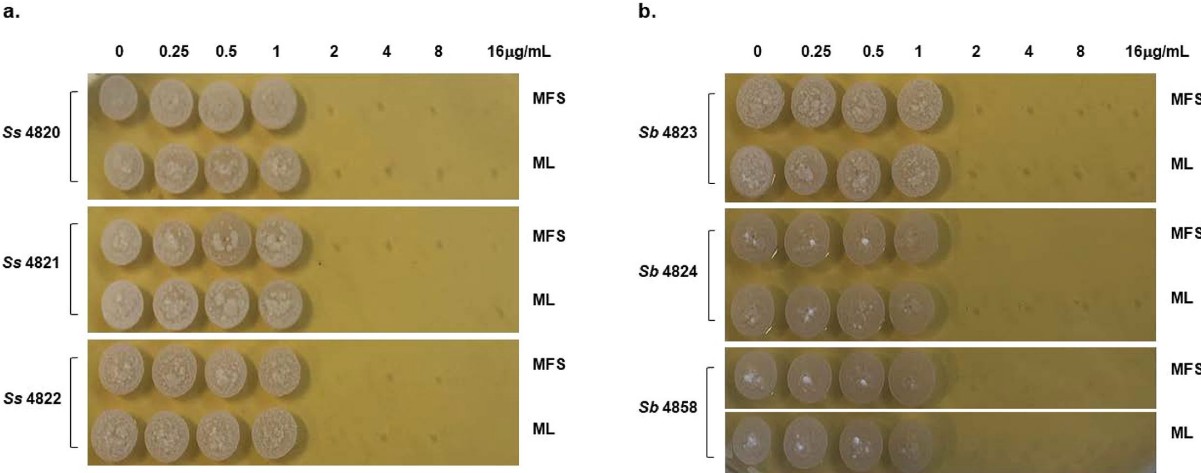

**FIG 1** *In vitro* fungicidal activity of miltefosine and milteforan against the yeast morphology of *S. schenckii* and *S. brasiliensis*. (a) *S. schenckii* (strains 4820, 4821, and 4822) and (b) *S. brasiliensis* (strains 4823, 4824, and 4858) yeast were grown in liquid YDP pH 7.8 at 37°C in the presence of several concentrations of MFS or ML (16, 8, 4, 2, 1, 0.5, and 0.25 µg/mL). After 4 days of incubation, the cells were plated in solid YPD pH 7.8 and incubated for 4 days at 37°C. As a control, yeast cells of each strain were grown without the drugs. The results represent the average of three independent experiments performed by duplicate.

(A549-yeast), and we observed a significant reduction of more than 90% in the fungal viability in both ML treatments, which contrasts with TRB treatment that shows about 50% viability (Fig. 5b).

When we challenged BMDMs with *S. brasiliensis* at a 1:10 ratio (BMDMs yeast) in the presence of 20 and 40 µg/mL ML, we observed complete clearing of *S. brasiliensis* compared with TRB that showed about 80% and 40% clearing, respectively, at 24 and 48 h (Fig. 6). Our results strongly indicated that ML can help both A549 and BMDMs to clear *S. brasiliensis* infection.

We also assessed the ability of the BMDMs to produce cytokines after stimulation by *S. brasiliensis* and treatment with the drug. It has already been reported that *S. brasiliensis* yeast stimulates higher production of TNF-α, IL-6, IL-1β, and IL-10 in human monocyte-derived macrophages when compared with *S. schenckii*, and it is also more phagocytosed (55), which might contribute to the higher virulence of this species.

After infection of BMDMs and treatment during 24 h, we observed a significant decrease in the stimulation of TNF-α and IL-6 when the yeast cells were treated with TRB and 20 and 40 µg/mL of ML, when compared with untreated cells (1:10) (Fig. 7a). However, when compared with TRB treatment, a significant decrease was observed in the stimulation of TNF-α only at 40 µg/mL of ML. In contrast, no difference was observed in the case of IL-6 with both ML concentrations compared with TRB. Finally, for the secretion of IL-10, a significant decrease was only observed when the yeast cells were treated with both ML concentrations. However, no difference was found between the TRB treatment and untreated cells (Fig. 7a).

After 48 h of infection, treatment with TRB did not cause a significant decrease in the TNF-α production, whereas both ML concentrations did when compared with untreated cells and TRB treatment (Fig. 7b). In the case of the IL-6 secretion, the same trend as that of 24 h was observed, with the only exception that treatment with 20 and 40 µg/mL of ML results in a significant decrease compared with TRB (Fig. 7b). The secretion of IL-10 did not decrease with the TRB treatment while significantly decreased in macrophages infected and uninfected treated with ML, confirming the participation of this drug in the immune response modulation (Fig. 7b).

**TABLE 2** MIC and MFC values of MFS and ML combination with several antifungals against *S. schenckii* and *S. brasiliensis* yeast phase[a]

| | | | CSP (16–0.25 µg/mL) | VCZ (16–0.25 µg/mL) | ITZ (8–0.125 µg/mL) | TRB (4–0.06 µg/mL) | AMB (8–0.125 µg/mL) |
|---|---|---|---|---|---|---|---|
| Ss 4820 | Y | MIC | >16 | >16 | 0.25 | 1 | 2 |
| | | MFC | >16 | >16 | 0.5 | 1 | 2 |
| | ML | MIC | >16 | >16 | <0.125 | <0.06 | 1 |
| | | MFC | >16 | >16 | 0.5 | <0.06 | 1 |
| | MFS | MIC | >16 | >16 | <0.125 | <0.06 | 1 |
| | | MFC | >16 | >16 | 0.5 | <0.06 | 1 |
| Ss 4821 | Y | MIC | >16 | >16 | 0.5 | 0.5 | 2 |
| | | MFC | >16 | >16 | 2 | 0.5 | 2 |
| | ML | MIC | >16 | >16 | <0.125 | 0.25 | 0.5 |
| | | MFC | >16 | >16 | 0.5 | 0.25 | 0.5 |
| | MFS | MIC | >16 | >16 | <0.125 | 0.25 | 0.5 |
| | | MFC | >16 | >16 | 0.5 | 0.25 | 0.5 |
| Ss 4822 | Y | MIC | >16 | >16 | 0.125 | 0.5 | 2 |
| | | MFC | >16 | >16 | 0.25 | 0.5 | 2 |
| | ML | MIC | >16 | >16 | <0.125 | 0.25 | 1 |
| | | MFC | >16 | >16 | 0.5 | 0.25 | 1 |
| | MFS | MIC | >16 | >16 | <0.125 | 0.25 | 1 |
| | | MFC | >16 | >16 | 0.5 | 0.25 | 1 |
| Sb 4823 | Y | MIC | >16 | >16 | 2 | 0.5 | 2 |
| | | MFC | >16 | >16 | >8 | 0.5 | 2 |
| | ML | MIC | >16 | >16 | 0.5 | 0.125 | 0.5 |
| | | MFC | >16 | >16 | >8 | 0.125 | 0.5 |
| | MFS | MIC | >16 | >16 | 0.5 | 0.125 | 0.5 |
| | | MFC | >16 | >16 | >8 | 0.125 | 0.5 |
| Sb 4824 | Y | MIC | >16 | >16 | 0.5 | 0.5 | >8 |
| | | MFC | >16 | >16 | 1 | 0.5 | >8 |
| | ML | MIC | >16 | >16 | 0.25 | 0.25 | 8 |
| | | MFC | >16 | >16 | 0.25 | 0.25 | 8 |
| | MFS | MIC | >16 | >16 | 0.25 | 0.25 | 8 |
| | | MFC | >16 | >16 | 0.25 | 0.25 | 8 |
| Sb 4858 | Y | MIC | >16 | >16 | 0.125 | 0.125 | 2 |
| | | MFC | >16 | >16 | 2 | 0.125 | 2 |
| | ML | MIC | >16 | >16 | <0.125 | 0.25 | 0.25 |
| | | MFC | >16 | >16 | 0.5 | 0.25 | 0.25 |
| | MFS | MIC | >16 | >16 | <0.125 | 0.25 | 0.25 |
| | | MFC | >16 | >16 | 0.5 | 0.25 | 0.25 |

[a]*Ss*: *S. schenckii*, *Sb*: *S. brasiliensis*; Y: untreated yeasts, ML: yeast treated with milteforan (1 µg/mL), MFS: yeast treated with miltefosine (1µg/mL); CSP: caspofungin, VCZ: voriconazole, ITZ: itraconazole, TRB: terbinafine, AMB: amphotericin B.

## DISCUSSION

Although there are several therapeutic options for the treatment of sporotrichosis, fungal resistance and cytotoxicity of the drugs to the host are essential obstacles that hinder the efficient recovery of the patient. ITZ is considered the first-line treatment, an azole known for its fungistatic activity against *Sporothrix* species (22, 24), which has increased the development of resistance in some isolates, mainly from *S. brasiliensis*, the most virulent species in the clinical clade (46, 56, 57). TRB, a drug with fungicidal activity against *Sporothrix,* has been reported to be effective in treating the cutaneous forms but not for the disseminated infections, for which AMB is used. AMB is considered a second-line treatment and is commonly used to treat the invasive and disseminated forms, with the disadvantage that it is very toxic in the doses and time needed to eradicate the infection, in addition to recent reports of isolates resistant to this antifungal agent (22, 46). Additional important variables to take under consideration when trying to establish

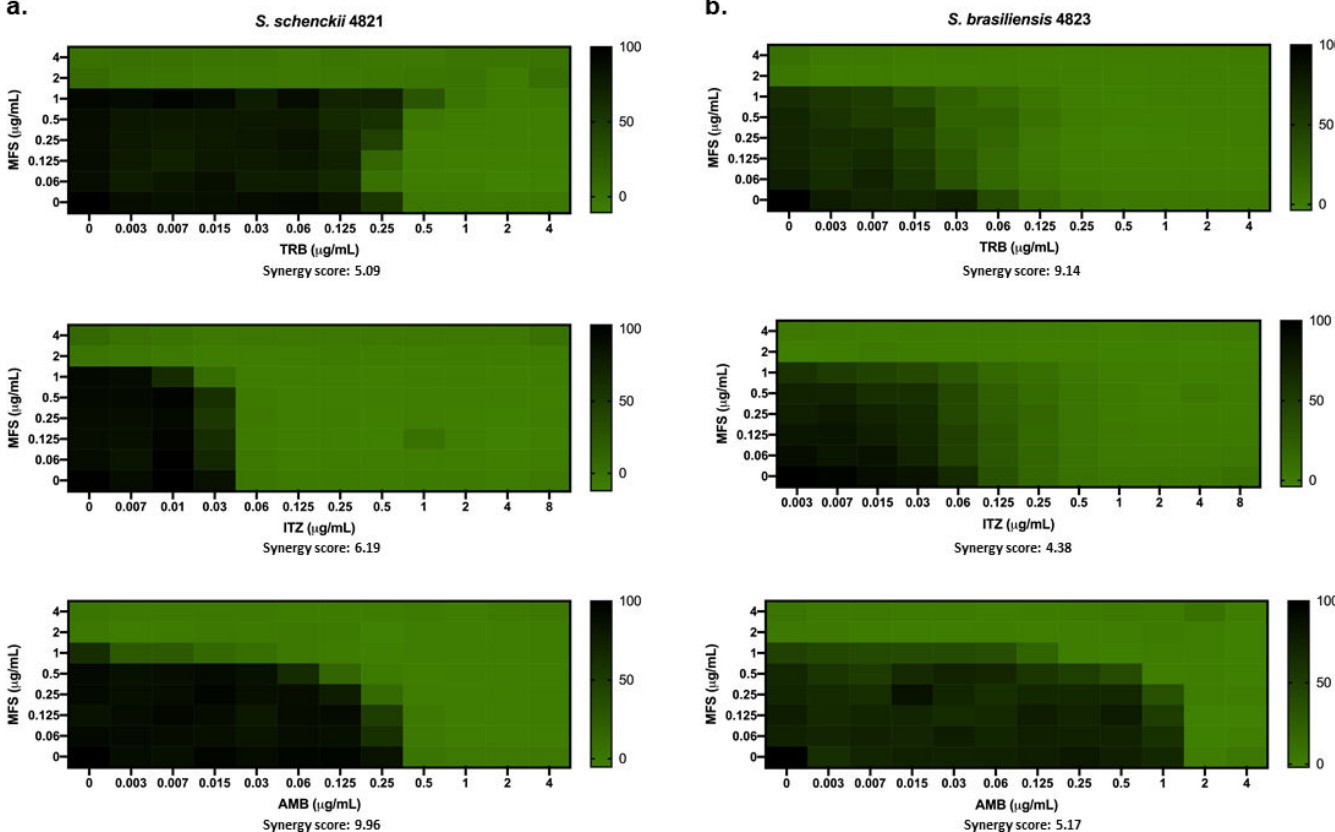

**FIG 2** MFS has an additive interaction with ITZ, TRB, and AMB against *S. brasiliensis* and *S. schenckii* yeast cells. The synergy score for MFS × TRB, MFS × ITZ, and MFS × AMB against *Sporothrix* was determined by analyzing the checkerboard assay data with SynergyFinder software. (a) *S. schenckii* and (b) *S. brasiliensis* yeasts were grown in liquid YDP pH 7.8 at 37°C in different concentrations of the selected drugs. After 4 days of incubation, the metabolic activity of the cells was assessed by the XTT reduction assay. The results are expressed as the % of metabolic activity and represent the average of three independent experiments.

a protocol for the sporotrichosis treatment, include (i) the etiological agent causing the infection, due to differences in the virulence profiles and the antifungal susceptibilities between the *Sporothrix* species; (ii) the clinical form affecting the patient; (iii) the immune status of the host, since this mycosis can cause disseminated fatal infections in immunocompromised patients; and (iv) the site of inoculation and the inoculum size (58). Sporotrichosis is a challenging mycosis to treat due to the aforementioned factors, including the difficulty of rapid diagnosis. The emergence of drug resistance and medication toxicity further complicate treatment, especially in endemic areas like Brazil, and these challenges are becoming a growing concern in other countries.

In Brazil, cat-transmitted sporotrichosis due to *S. brasiliensis* has been a critical health threat since 1998 (5, 8), spreading across the country and affecting both humans and domestic animals such as cats and dogs. This ongoing outbreak underscores the urgent need for new drugs to treat and control this mycosis. One promising approach is drug repurposing, which involves using existing drugs already approved for treating other diseases to treat new infections. This method has been proposed as an excellent alternative for finding new therapies. For instance, commercial MFS, developed initially as an antineoplastic drug (27, 28), is now the only available oral treatment for leishmaniasis in dogs and humans (29–32). Recent studies have shown its effectiveness against *Candida* species (39, 40). As previously demonstrated (45, 46, 48), we also found that MFS exhibits *in vitro* fungicidal activity against *Sporothrix* species by inhibiting the growth of both fungal morphologies. Strains of *S. brasiliensis* and *S. schenckii* were sensitive to low concentrations of this drug, with MIC and MFC values of 2 µg/mL for both hyphae and yeast-like cells. Unlike ITZ, no strain resistant to MFS or ML was found in our studies.

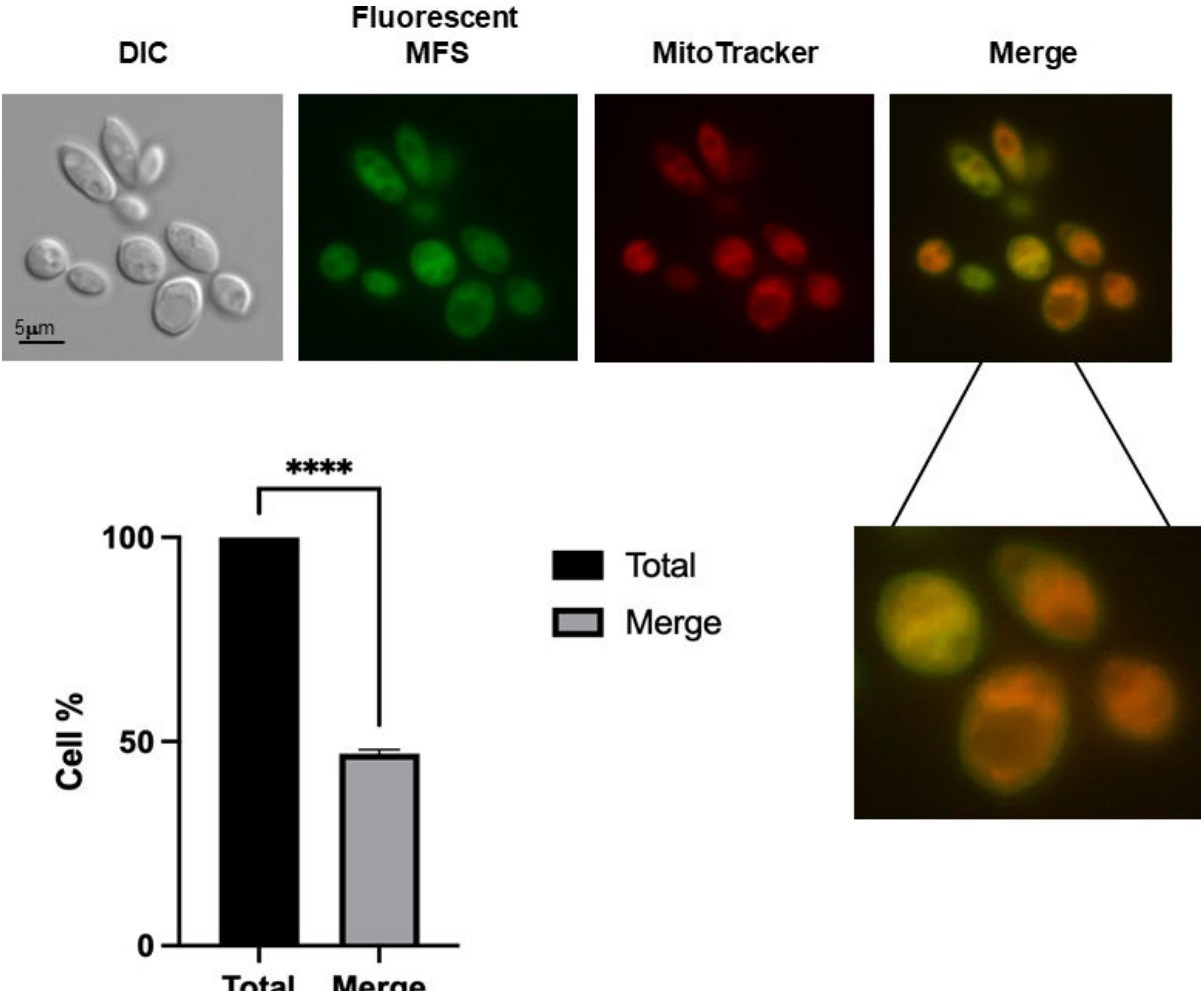

**FIG 3** MFS is localized in the mitochondria and cell surface of *S. brasiliensis* yeast. *S. brasiliensis* yeast cells were exposed to fluorescent MFS (2 µg/mL) for 1 h and then stained with MitoTracker Deep Red FM. The MFS and MitoTracker signals merged on the mitochondria, whereas the MFS signal was observed on the cell surface. Three independent experiments were performed, and 100 cells were counted for each to calculate a 47.06% ± 1.01% of MFS and MitoTracker colocalization (merge).

We also assessed the ability of MFS to synergize with other drugs used for the treatment of sporotrichosis, including TRB, ITZ, and AMB, and as previously reported for ITZ (45), MFS does not synergize the activity of these antifungals. However, it has an additive effect, suggesting that they do not interact on independent pathways (59).

Although these results *in vitro* could suggest a positive clinical outcome in the treatment of sporotrichosis with MFS, there is a lack of studies associating the *in vitro* susceptibility of the *Sporothrix* species with the *in vivo* therapeutic response (14). Actually, no association between lower MIC values of AMB and ITR with a higher cure rate was found, which might suggest that other factors, such as the immune status of the patient, the degree of dissemination, the early start of the treatment, and the antifungal absorption in the host, are as important as antifungal susceptibility (60, 61). Therefore, the *in vitro* findings could differ from the *in vivo* outcome. An example of this was observed when the effectiveness of MFS was evaluated in cats with refractory sporotrichosis (62). Among 10 cats that had been treated with ITZ or potassium iodide for over a year, one cat received MFS (2 mg/kg orally every 24 h) for 45 days, 6 for 30 days, 1 for 21 days, 1 for 15 days, and 1 for 3 days. Most of the cats treated with MFS showed disease progression, along with hyporexia and weight loss, suggesting MFS treatment failure (62). However, further studies with a larger number of patients and

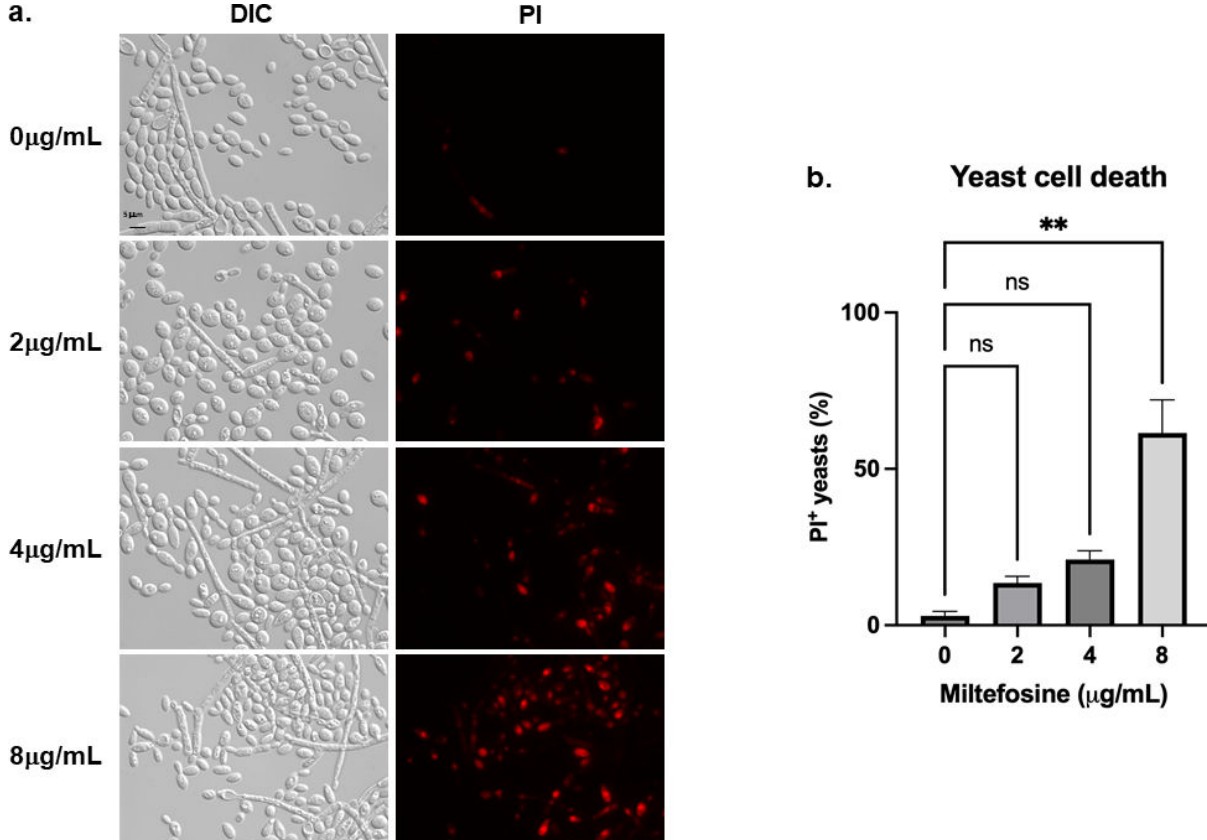

**FIG 4** MFS causes dose-dependent death in *S. brasiliensis* yeast. (a) *S. brasiliensis* yeasts were exposed to 0, 2, 4, and 8 µg/mL of MFS for 6 h, stained with PI, and analyzed by fluorescence microscopy. (b) Quantification of PI⁺ yeast exposed to MFS, in which 100 yeast-like cells were counted for each condition. The results represent the average of two independent experiments. \*\**P* value < 0.001 when compared with untreated cells; ns: not significant.

testing different MFS concentrations and treatment duration are needed to assess the MFS efficacy *in vivo*.

In fungi, the MFS mechanism of action is still poorly understood, but it was found in *Saccharomyces cerevisiae* that the drug uptake is fast, penetrating the mitochondrial inner membrane and disrupting its potential, which eventually leads to apoptosis. One of the MFS potential targets was identified, *COX9*, which encodes a subunit of the cytochrome c oxidase complex in the electron transport chain of the mitochondrial membrane (63). An increase in plasma membrane permeability, a decrease in mitochondrial membrane potential, and an increase in ROS production after treatment with MFS were also observed in *Cryptococcus neoformans* and *Scedosporium aurantiacum* (41, 64). Interestingly, *Scedosporium* genera and *Sporothrix* species are the only medically relevant fungi reported to present rhamnose in their cell walls (65).

In *S. brasiliensis* yeasts, it was previously found that MFS alters the plasma membrane integrity, decreases the cell wall thickness, increases the microfibrillar layer (peptidorhamnomannan) thickness, and increases the melanin content in the cell wall (24). These results explain the localization of MFS in *S. brasiliensis* yeasts mitochondria and cell surface that we observed, confirming that MFS mechanism of action is related to the mitochondria and cell membrane integrity and suggesting that this is indeed a multi-target drug.

MFS and ML have been reported to be toxic at high doses in mice, with high mortality in concentrations higher than 25 mg/kg (66, 67), with maximum concentrations of the drug in the kidney and liver, probably due to its amphiphilic nature (68, 69). Therefore, we assessed ML cytotoxicity in A549 human pulmonary cells and observed a significant

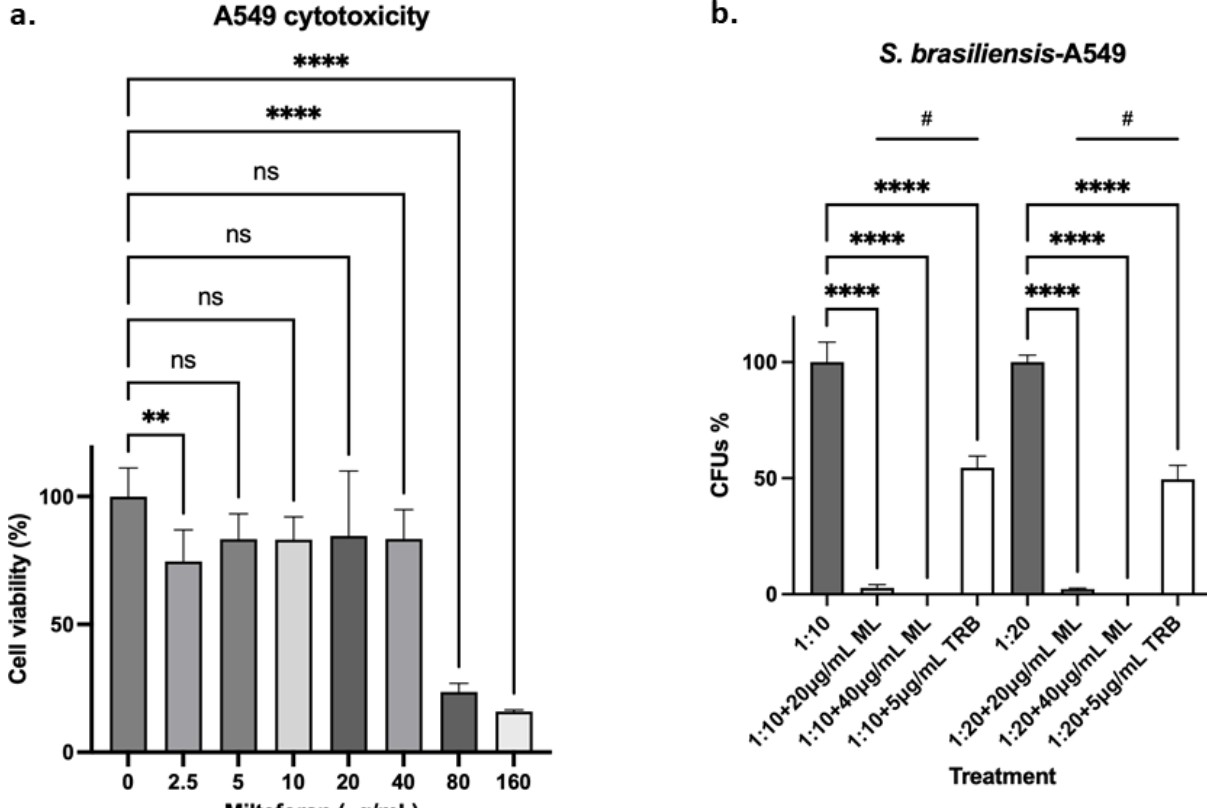

**FIG 5** Concentrations from 5 to 40 µg/mL of ML are not toxic to human cells and can significantly decrease *S. brasiliensis* survival in A549 epithelial cells. (a) A459 epithelial cells were treated with different ML concentrations for 48 h, with a decrease of cell viability only at 80 µg/mL or higher concentrations. (b) A459 cells were challenged with *S. brasiliensis* yeasts at a proportion of 1:10 and 1:20 and then treated with 20 and 40 µg/mL of ML for 24 h. The fungicidal drug TRB was included as a control. **$P$-value < 0.01 when compared with untreated cells; ****$P$-value < 0.0001 when compared with untreated cells; #$P$-value < 0.0001 when compared with cells treated with TRB.

viability reduction at 80 µg/mL. When we tested the ability of ML to decrease the fungal burden in A549 cells and BMDMs, at 24 h, and 24 and 48 h, respectively, we observed that ML could significantly reduce the CFUs more efficiently than the fungicidal drug TRB in both cell types, with an almost complete clearing of the yeast cells as early as 24 h of treatment. These results show that ML is capable of killing the parasitic morphology of *S. brasiliensis* in the host tissue at a concentration high enough to exert fungicidal activity, yet low enough to avoid toxicity to host cells.

Another proposed mechanism of action for MFS is its immunomodulatory ability, which is essential for treating leishmaniasis. MFS induces the Th1 response and suppresses the Th2 response by increasing the production of proinflammatory cytokines such as IFNγ, TNFα, and IL-12, aiding in the clearance of intracellular pathogens. This immunomodulation is vital to avoid relapses of leishmaniasis, which are associated with an increased Th2 response and higher IL-10 production (32, 38). Here, we observed that ML decreases the fungal burden and the production of TNFα, IL-6, and IL-10 in infected BMDMs. Thus, we propose three non-excluded hypotheses to explain this immune response: (i) the reduction in cytokines might be related to the drug killing the yeasts before they are phagocytosed, as we observed MFS-induced yeast cell death as early as six hours; (ii) MFS, localized to the cell surface, might act as an opsonizing agent, enhancing macrophage recognition and subsequent phagocytosis; and (iii) MFS could bind to essential virulence factors, such as adhesins, or to immunogenic cell wall components like β-glucans. This binding may attenuate *S. brasiliensis'* ability to infect and elicit a robust immune response, facilitating fungal clearance. All three mechanisms

## a. *S. brasiliensis*-BMDM 24h

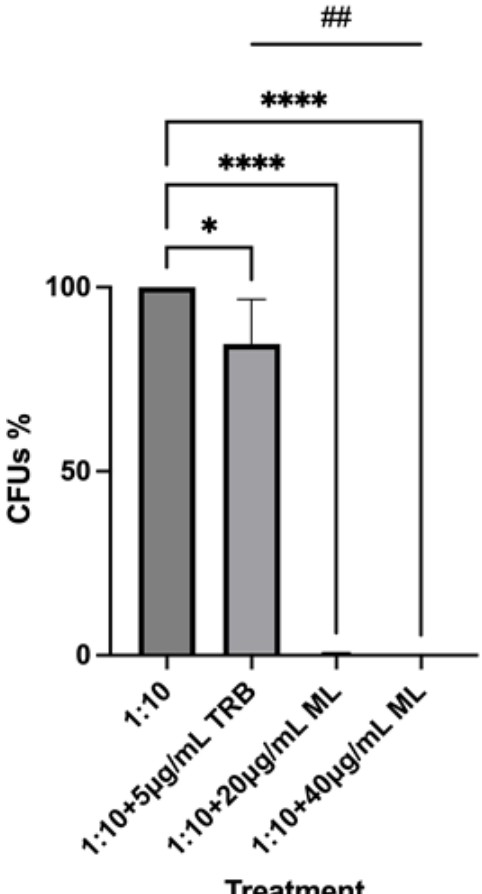

## b. *S. brasiliensis*-BMDM 48h

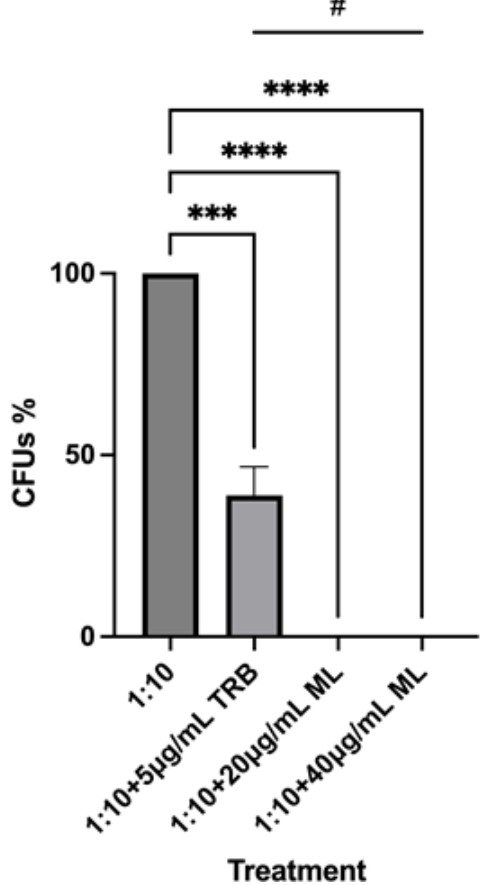

**FIG 6** The killing of *S. brasiliensis* yeasts by BMDM significantly increases in the presence of ML. (a) BMDM cells were infected with *S. brasiliensis* yeasts and then treated with 20 and 40 µg/mL of ML for 24 h, which decreased the fungal survival by almost 100% when compared with untreated cells. (b) BMDM cells were infected with *S. brasiliensis* yeasts and were then treated with 20 and 40 µg/mL of ML for 48 h, which decreased the fungal survival to 100% when compared with untreated cells. The fungicidal drug TRB was included as a control. *$P$-value < 0.05 when compared with untreated cells; ***$P$-value < 0.0005 when compared with untreated cells; ****$P$-value < 0.0001 when compared with untreated cells; #$P$-value < 0.01 when compared with cells treated with TRB; ##$P$-value < 0.01 when compared with cells treated with TRB.

would reduce the fungal load, tissue damage, and inflammation, thereby decreasing *Sporothrix* virulence and positioning MFS as a promising alternative treatment for feline sporotrichosis.

As previously mentioned, the *in vitro* response of an antifungal against *Sporothrix* may not accurately predict treatment outcomes in an *in vivo* model, and this has been observed with these pathogenic fungi. Therefore, we only suggest MFS as a potential candidate for treating sporotrichosis in cats, the primary vector for this mycosis transmission. However, the optimal dosage, treatment duration, and specific characteristics of the host need further evaluation to determine the efficacy of MFS for feline sporotrichosis treatment.

## MATERIALS AND METHODS

### Fungal strains and culture conditions

In this study, three *Sporothrix schenckii* (ATCC-MYA 4820, ATCC-MYA 4821, and ATCC-MYA 4822) and three *S. brasiliensis* strains (ATCC-MYA 4823, ATCC-MYA 4824, and ATCC-MYA

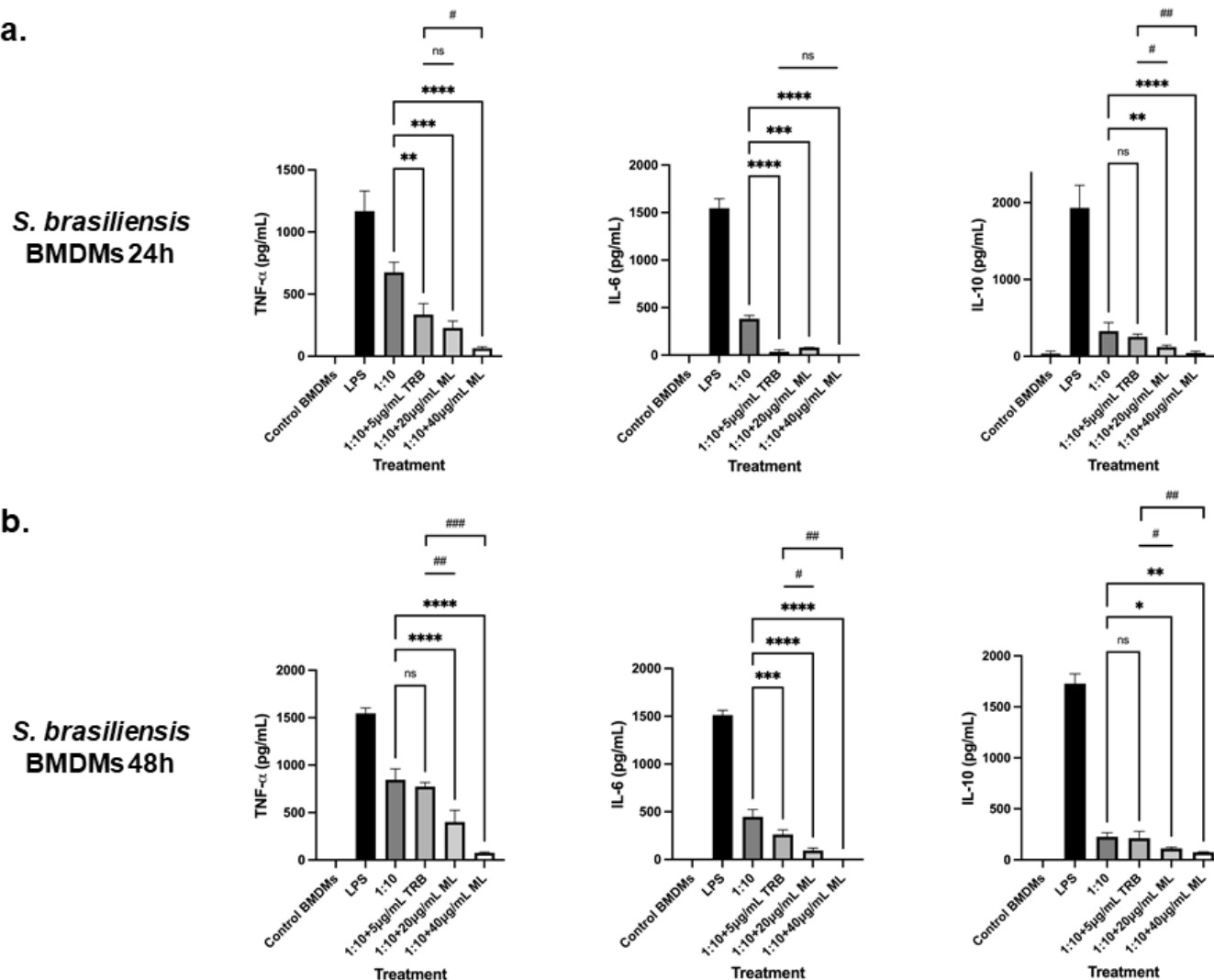

**FIG 7** Cytokine secretion by BMDM infected with *S. brasiliensis* and treated with ML. (a) BMDM cells were infected with *S. brasiliensis* yeasts and treated with 20 and 40 µg/mL of ML for 24 h. The interaction supernatant was collected and the cytokines TNF-α (ns: not significant; **P-value < 0.005 when compared with untreated cells; ***P-value < 0.0005 when compared with untreated cells; ****P-value < 0.0001 when compared with untreated cells; #P-value < 0.01 when compared with TRB treatment), IL-6 (ns: not significant; ***P-value < 0.0005 when compared with untreated cells; ****P-value < 0.0001 when compared with untreated cells), and IL-10 (ns: not significant; **P-value < 0.005 when compared with untreated cells; ****P-value < 0.0001 when compared with untreated cells; #P-value < 0.0005 when compared with TRB treatment; ##P-value < 0.0001 when compared with TRB treatment) were measured. (b) BMDM cells were infected with *S. brasiliensis* yeasts and treated with 20 and 40 µg/mL of ML for 48 h. The interaction supernatant was collected and the cytokines TNF-α (ns: not significant; ****P-value < 0.0001 when compared with untreated cells; #P-value < 0.0005 when compared with TRB treatment; ##P-value < 0.0001 when compared with TRB treatment), IL-6 (***P-value < 0.001 when compared with untreated cells; ****P-value < 0.0001 when compared with untreated cells; #P-value < 0.005 when compared with TRB treatment; ##P-value < 0.0001 when compared with TRB treatment), and IL-10 (ns: not significant; *P-value < 0.05 when compared with untreated cells; **P-value < 0.01 when compared with untreated cells; #P-value < 0.05 when compared with TRB treatment; ##P-value < 0.01 when compared with TRB treatment) were measured.

4858) were used for the *in vitro* antifungal susceptibility assays; *S. schenckii* ATCC-MYA 4821 and *S. brasiliensis* ATCC- MYA 4823 were used for the checkerboard assays; and *S. brasiliensis* ATCC-MYA 4823, a highly virulent clinical isolate obtained from feline sporotrichosis (70), was used for the infection assays.

The mycelial phase from *Sporothrix* spp. was obtained and maintained on solid YPD pH 4.5 [yeast extract 1% (wt/vol), gelatin peptone 2% (wt/vol), and dextrose 3% (wt/vol)] at 28°C for 4 days. In contrast, the yeast morphology was grown in liquid YPD pH 7.8, at 37°C under orbital agitation for 4 days, as previously reported (71). For the experiments

in which we needed a single morphology, the cultures were filtered with sterile miracloth (Calbiochem) to avoid contamination with the unwanted fungal morphotype. Each phase was confirmed by observing the cells with light microscopy.

## Antifungal drugs

For the *in vitro* assays, voriconazole (VCZ, Sigma-Aldrich), itraconazole (ITZ, Sigma-Aldrich), amphotericin B (AMB, Sigma-Aldrich), terbinafine (TRB, Sigma-Aldrich), and brilacidin (BRI, supplied by Innovation Pharmaceuticals) were diluted in dimethyl sulfoxide (DMOS); while miltefosine (MFS, Sigma-Aldrich), the milteforan active compound, was diluted in ethanol; and caspofungin (CSP, Sigma-Aldrich) was diluted in distilled water. Milteforan (miltefosine 2%) was purchased from Virbac as an oral solution.

## *In vitro* antifungal susceptibility testing

The minimum inhibitory concentrations (MICs) were determined by the broth microdilution method adapted from protocols published by the Clinical Laboratory Standard Institute for the mycelial and yeast phases (24, 72). Briefly, serial two-fold dilutions of the antifungal drugs were performed in YPD pH 4.5 and 7.8, for mycelial and yeast, respectively, into 96-well microtiter plates to obtain concentrations of 4–0.06 µg/mL for CSP, VCZ, and TRB; 8–0.125 µg/mL for ITZ and AMB; 16–0.25 µg/mL for MFS and ML; and 80–1.25 µM for BRI, with a final concentration of $2 \times 10^3$ and $2 \times 10^4$ conidia or yeast cells, respectively, in a volume of 100 µL. The plates were incubated at 28°C (for conidia) or 37°C (for yeast) for 4 days, and the MIC was determined by visual inspection and defined as the lowest concentration that inhibits 90-100% of fungal growth about untreated cells. Finally, 5 µL of conidia or yeast cells from each well were grown in drug-free solid YPD pH 4.5 and pH 7.8 at 28°C and 37°C, respectively, for 4 days. The minimum fungicidal concentration (MFC) value was the lowest concentration, showing no fungal growth. Three independent experiments were performed by duplicate.

## Checkerboard assays and synergy testing

The drug combination effect was determined through the MIC and MFC values of the yeast phase, as described before. Briefly, serial twofold dilutions of the antifungal drugs were performed in liquid YPD pH 7.8 containing half MIC of MFS or ML (1 µg/mL) in 96-well microtiter plates to obtain concentrations of 16–0.25 µg/mL for CSP and VCZ; 8–0.125 µg/mL for ITZ and AMB; 4–0.06 µg/mL for TRB; and 80–1.25 µM for BRI, with a final concentration of $2 \times 10^4$ yeast, in a volume of 100 µL. The plates were incubated at 37°C for 4 days, and the MIC was determined by visual inspection. MIC was defined as the lowest concentration inhibiting 90%–100% of fungal growth in cells treated only with 1 µg/mL of MFS or ML. After MIC determination, 5 µL of yeast from each well were grown in drug-free solid YPD pH 7.8 at 37°C for 4 days. The MFC value was the lowest concentration, which showed no fungal growth.

Checkerboard assays were performed to quantify the interaction (synergistic, additive, or antagonistic) between MFS and ITZ, AMB, or TRB. Briefly, a stock solution of $2 \times 10^5$ yeast/mL and each drug (8 µg of MFS and 16 µg/mL of ITZ, 16 µg of AMB, or 8 µg of TRB) were prepared in RMPI-1640. In 96-well microtiter plates, the first antibiotic (MFS) was diluted sequentially along the ordinate. In contrast, the second drug (ITZ, AMB, or TRB) was diluted along the abscissa to obtain a final volume of 100 µL. The plates were incubated at 37°C for 4 days, and the metabolic activity was determined through the XTT reduction assay (47). Briefly, 50 µL of a solution of XTT 1 mg/mL and menadione 1 mM resuspended in water were added to each well, mixed, and incubated in the dark at 37°C for three h. The supernatant of each well was transferred to a new plate and read in a spectrophotometer at 492 nm. Results are expressed as means ± SD of three independent experiments.

The SynergyFinder software (52) was used to determine the type of drug interaction with the following parameters: detect outliners: yes; curve fitting: LL4; method: Bliss;

correction: on. The summary synergy scores represent the average excess response due to drug interaction, in which a value less than −10 suggests an antagonistic interaction between two drugs; values from −10 to 10 suggest an additive interaction; and values larger than 10 suggest a synergistic interaction.

## Yeast cells death

The effect of ML on the cell membrane potential was assessed by staining with propidium iodide (PI). Yeast cells grown for 4 days in liquid YPD pH 7.8 were treated with 0, 2, 4, and 8 µg/mL of ML for 6 h, stained with PI 20 mM for 30 minutes, and washed with PBS 1 × three times. Fluorescence was analyzed at an excitation wavelength of 572/25 nm and emission of 629/62 nm with the Observer Z1 fluorescence microscope using a 100 × oil immersion lens objective. Differential interference contrast (DIC) and fluorescent images were captured with an AxioCam camera (Carl Zeiss) and processed using AxioVision software (version 4.8). The experiment was performed twice, and at least 100 cells were counted for each treatment. The results were plotted using GraphPad Prism (GraphPad Software, Inc). A $P$-value < 0.001 was considered significant.

## Miltefosine localization

*S. brasiliensis* yeast cells cultured for 4 days in YPD pH 7.8 were washed three times with PBS 1 × and then treated with the fluorescent MFS analog MT-11 C-BDP (excitation wavelength 450–490nm and emission wavelength 500–550nm) for 6 hours, also in liquid YPD pH 7.8. The cells were washed three times, stained with 250 nM of MitoTracker Deep Red FM (Invitrogen) (wavelength absorbance/emission 644/665 nm), and washed again. The yeast cells were visualized in slides with the Observer Z1 fluorescent microscope using a 100 x oil immersion lens objective. DIC and fluorescent images were captured with an AxioCam camera (Carl Zeiss) and processed using AxioVision software (version 4.8). Two independent experiments were performed, and 100 cells were counted each to calculate the merge %.

## Cytotoxicity assay

The cytotoxicity of ML was determined in A549 human lung cancer cells using the XTT reduction assay. A total of $2 \times 10^5$ cells/well were seeded in 96-well tissue plates and incubated in Dulbecco's Modified Eagle Medium (DMEM, Thermo Fischer). After 24 h of incubation with $CO_2$ 5%, the cells were treated with different concentrations of ML (0, 2.5, 5, 10, 20, 40, 80, and 160 µg/mL), and after 48 h of incubation, cell viability was assessed using the XTT assay. Briefly, 80 µL of a solution of XTT 1 mg/mL in DMEM, HEPES 1 M, and menadione 8 µg/mL were added to each well, and after 30 min, formazan formation was quantified spectrophotometrically at 450 nm using a microplate reader. Each treatment was performed in triplicate, and the results were plotted using GraphPad Prism (GraphPad Software, Inc). A $P$-value < 0.0001 was considered significant.

## A549 and bone marrow-derived macrophages (BMDMs) killing assays

The cytotoxicity of ML was determined in A549 human lung cancer cells [ATCC, CCL-185, derived from Rio de Janeiro Cell Bank, Brazil (BCRJ-0033) passage (5–10)] using the XTT reduction assay. The cell line A549 and BMDMs were cultured using DMEM supplemented with fetal bovine serum (FBS) 10% and penicillin-streptomycin 1% (Sigma-Aldrich) and seeded at a concentration of $1 \times 10^6$ cells/mL in 24-well plates (Corning). The cells were challenged with *S. brasiliensis* yeasts at a multiplicity of infection of 1:10 and were then treated with ML 20 and 40 µM. We included untreated cells and cells treated with TRB 5 (g/mL) as a control. For the BMDMs, cells treated with LPS were also included as controls. The A549 were incubated for 24 h at 37°C with $CO_2$ 5%, while the BMDM were incubated for 24 and 48 h under the same conditions. After incubation, the culture media was removed, each well was washed three times with PBS 1×, and 1 mL of sterile cold water was added to recover and collect the cell monolayer. To assess the number of

CFUs, 100 µL of the cell suspensions were plated on YDP pH 4.5 and incubated at 28°C for 4 days. When necessary, the cell suspensions were diluted at 1:100 or 1:1000, and 100 µL were plated. 50 µL of the inoculum adjusted to $1 \times 10^3$ cells/mL was also plated to correct the CFU count. Each treatment was performed in triplicate to calculate the CFU %, and the results were plotted using GraphPad Prism (GraphPad Software, Inc). A *P*-value < 0.0001 was considered significant.

## Cytokines quantification

The Elisa-assay kits (R&D Systems) were used to evaluate the concentration of the proinflammatory cytokines TNFα and IL-6, and the anti-inflammatory cytokine IL-10 in the supernatants of the *S. brasiliensis* and BMDMs interaction for 24 and 48 h, according to the manufacturer's instruction. The plate's absorbance was read at 450 nm, and the cytokine concentration (pg/mL) was calculated according to the values obtained in the standard curve of each cytokine. The results were plotted using GraphPad Prism (GraphPad Software, Inc).

## Statistical analyses

The GraphPad Prism 10 (GraphPad Software, Inc.) was used for the statistical analyses. The results are reported as the media ± SD from two or three independent experiments performed by duplicate and were analyzed using the Ordinary one-way ANOVA or the Unpaired *t*-test. The statistical significance was considered with a *P*-value < 0.05 or lower.

## ACKNOWLEDGMENTS

We thank the Fundação de Amparo à Pesquisa do Estado de São Paulo (FAPESP) grant numbers 2021/04977–5 (GHG) and 2022/08556–7 (LCGC) 2022/08796–8 (CD), 2022/09882–5 (LP), the Conselho Nacional de Desenvolvimento Científico e Tecnológico (CNPq) and Fundação Coordenação de Aperfeiçoamento do Pessoal do Ensino Superior (CAPES) grant number 405934/2022–0 (The National Institute of Science and Technology INCT Funvir), and CNPq 301058/2019–9 from Brazil to GHG., both from Brazil, the National Institutes of Health/National Institute of Allergy and Infectious Diseases from the USA, grant R01AI153356 to GHG. This work was also funded by the Joint Canada-Israel Health Research Program, jointly supported by the Azrieli Foundation, Canada's International Development Research Centre, Canadian Institutes of Health Research, and the Israel Science Foundation (GHG).

## AUTHOR AFFILIATIONS

[1]Faculdade de Ciências Farmacêuticas de Ribeirão Preto, Universidade de São Paulo, Ribeirão Preto, Brazil
[2]Department of Microbiology, Immunology and Parasitology, Discipline of Cellular Biology, Laboratory of Emerging Fungal Pathogens, Federal University of São Paulo, São Paulo, Brazil
[3]National Institute of Science and Technology in Human Pathogenic Fungi, São Paulo, Brazil

## AUTHOR ORCIDs

Camila Diehl ⓘ https://orcid.org/0000-0003-3174-0346
Gustavo H. Goldman ⓘ http://orcid.org/0000-0002-2986-350X

## FUNDING

| Funder | Grant(s) | Author(s) |
| --- | --- | --- |
| Fundação de Amparo à Pesquisa do Estado de São Paulo (FAPESP) | 2022/08556-7 | Laura C. García Carnero |

| Funder | Grant(s) | Author(s) |
|---|---|---|
| Fundação de Amparo à Pesquisa do Estado de São Paulo (FAPESP) | 2022/09882-5 | Lais Pontes |
| Fundação de Amparo à Pesquisa do Estado de São Paulo (FAPESP) | 2022/08796-8 | Camila Diehl |
| Fundação de Amparo à Pesquisa do Estado de São Paulo (FAPESP) | 2021/04977-5 | Gustavo H. Goldman |

## AUTHOR CONTRIBUTIONS

Laura C. García Carnero, Data curation, Formal analysis, Investigation, Writing – original draft | Camila Figueiredo Pinzan, Investigation, Writing – review and editing | Camila Diehl, Investigation, Writing – review and editing | Patricia Alves de Castro, Investigation, Writing – review and editing | Lais Pontes, Investigation, Writing – review and editing | Anderson Messias Rodrigues, Resources, Writing – review and editing | Thaila F. dos Reis, Investigation | Gustavo H. Goldman, Conceptualization, Funding acquisition, Project administration, Writing – original draft, Writing – review and editing

## ADDITIONAL FILES

The following material is available online.

### Open Peer Review

**PEER REVIEW HISTORY (review-history.pdf).** An accounting of the reviewer comments and feedback.

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
