## [Reviewer comments · Microbiology Spectrum]

Microbiology Spectrum

Milteforan, a promising veterinary commercial product against feline sporotrichosis

Laura Carnero, Camila Pinzan, Camila Diehl, Patricia de Castro, Lais Pontes, Anderson Rodrigues, Thaila dos Reis, and Gustavo Goldman

Corresponding Author(s): Gustavo Goldman, Universidade de Sao Paulo

Review Timeline:

Submission Date:	February 19, 2024
Editorial Decision:	March 18, 2024
Revision Received:	July 7, 2024
Accepted:	July 12, 2024

Editor: Alexandre Alanio

Reviewer(s): Disclosure of reviewer identity is with reference to reviewer comments included in decision letter(s). The following individuals involved in review of your submission have agreed to reveal their identity: Luana Pereira Borba-Santos (Reviewer #1)

Transaction Report:

DOI: <https://doi.org/10.1128/spectrum.00474-24>

Re: Spectrum00474-24 (Milteforan, a promising veterinary commercial product against feline sporotrichosis)

Dear Dr. Gustavo H. Goldman:

Thank you for the privilege of reviewing your work. Below you will find my comments, instructions from the Spectrum editorial office, and the reviewer comments.

Revision Guidelines

Sincerely,
Alexandre Alanio
Editor
Microbiology Spectrum

Reviewer #1 (Comments for the Author):

The manuscript submitted by Carnero and colleagues presents a good exploration of the miltefosine activity against sporotrichosis agents. It is an attractive and well-structured manuscript that provides good insights into the understanding of the drug's potential for repurposing. Some minor suggestions for improvement are outlined below:

Line 157-158: I suggest retiring the results of brilacidin in this paper because the focus of the present study is to determine the

miltefosine effect.

Lines 216-228: The experiments with A549 and BMDM cells should be performed also using treatment with itraconazole as a reference antifungal.

Lines 256-258: The authors should mention that considering *Sporothrix* species, there is insufficient evidence between in vitro and clinical observations about antifungal activity.

Lines 290-291: The study Zuo et al 2011 (doi:10.1124/mol.111.072322) showed that miltefosine interacts with cytochrome c oxidase disrupting mitochondrial membrane potential in fungal cells, corroborating the localization of miltefosine observed in the present work. Besides, there are many other effects observed in fungal cells after miltefosine exposure described in the literature (doi: 10.1016/j.mycmed.2023.101436).

Lines 300-301: Why itraconazole was not included in these experiments?

Lines 319-320: The authors should mention that the activity of miltefosine in sporotrichosis treatments of cats was already evaluated and did not present good results in one study (doi: 10.22456/1679-9216.83639).

Line 321: I suggest including a paragraph with a conclusion about the study.

Reviewer #2 (Comments for the Author):

I have thoroughly reviewed the manuscript titled "Milteforan, a promising veterinary commercial product against feline sporotrichosis". While the manuscript offers valuable insights into the efficacy of Milteforan, there are several points that require the authors' attention to further strengthen the manuscript and enhance its overall quality.

1) To bolster the comprehensiveness of the study, I strongly recommend expanding the strain diversity, particularly for MIC and MFC analyses. Sporotrichosis is not a rare disease and it is completely feasible to increase *Sporothrix* strain numbers. Only three strains of each species is not acceptable. In addition, consider including strains with distinct genetic backgrounds, since it is reported that *S. brasiliensis* has considerable genetic diversity (see references with the following DOI: 10.1016/j.jinf.2023.02.034; 10.1093/mmy/myac096; 10.1016/j.fgb.2023.103845) that may change the results found by the authors.

2) It seems there might be a reference to Brilacidin that needs further clarification or elaboration in the manuscript. Brilacidin is presented in the results, but not mentioned in the abstract nor discussed. If brilacidin also has in vitro activity against *S. brasiliensis*, why the authors did not make more experiments with this drug as they did with miltefosine?

3) Why AMB Micelial MIC were not determined in Table 1? There appears to be a discrepancy regarding the determination of AMB Micelial MIC, which should be addressed for clarity.

4) Table 2 seems to have some entries marked as ND (not determined), which should be explained or resolved.

5) l. 209: shows, instead of showss.

6) In lines 208-210, authors say: "Treatment of *S. brasiliensis* yeasts with 2, 4, and 8 µg/mL of MFS showss dose-dependent damage of the cells since the PI signal increased with the drug concentration". However, as depicted in Figure 4b, only 8 µg/mL MFS significantly damaged cells. Please rewrite to make this clear.

7) Authors used lung cells and macrophages in their experiments. Sporotrichosis rarely affects the lungs. Consider diversifying the cell types used in your study to include skin cells, since sporotrichosis is a subcutaneous disease, and cat cells, especially considering the intended application for treating feline sporotrichosis, as mentioned in the title of the study.

8) There seems to be an issue in Fig 4a that warrants further clarification regarding the presence of hyphae amidst yeasts.

9) Authors claim in lines 218-219: "As shown in Figure 5a, ML concentrations of 40 µg/mL and lower did not reduce A549 cell viability compared to the control." Figure 5a shows that 2.5 µg/ml milteforan significantly reduced cell viability ($P < 0.01$). The statement in lines 218-219 regarding A549 cell viability should be corrected to reflect the significant reduction observed at 2.5 µg/ml of Milteforan, as shown in Figure 5a.

10) The comparison between ML and TRB treatments in Fig 5b requires further contextualization, considering the significant difference in concentrations used. ML concentration used is eight times higher than TRB concentration! Also, plasma

concentration of miltefosine (see PMID: 31685474) is 16 µg/ml after seven days of therapy (200 mg/day), which is 2.5 times lower than the concentration used in this experiment. The plasma concentration of Miltefosine should be considered in the interpretation of the results.

11) Figure 6 has the same issue with ML/TRB concentrations mentioned above. Moreover, the concentration units of ML need to be clarified for consistency between the figure and the text.

12) Assessment of cytokine production: While the assessment of BMDMs is intriguing, it would be more informative to include similar experiments using cat cells to provide a more comprehensive understanding, especially considering the differences in immunology across species. The immune response of cats against *S. brasiliensis* is different from that observed in humans and mice.

13) Similar to earlier points, the discrepancy in TRB/ML concentration in Fig 7 should be addressed for accuracy.

14) Discussion needs to be highly improved. The discussion section (64 lines) appears to be relatively shorter compared to the introduction (90 lines). It would be beneficial to expand upon the discussion to ensure a thorough exploration of the findings and their implications. Also, authors need to explore the limitations of their work and mention future studies needed to make Milteforan a promising veterinary commercial product against feline sporotrichosis, as stated in the title of the manuscript.

15) l. 440: Providing information regarding the origin and number of passages of the cell lines would enhance the transparency and reproducibility of the study.

Answers to the Reviewers

Reviewer #1:

The manuscript submitted by Carnero and colleagues presents a good exploration of the miltefosine activity against sporotrichosis agents. It is an attractive and well-structured manuscript that provides good insights into the understanding of the drug's potential for repurposing. Some minor suggestions for improvement are outlined below.

Answer: We thank the reviewer for the comments

1) Line 157-158: I suggest retiring the results of brilacidin in this paper because the focus of the present study is to determine the miltefosine effect.

Answer: These results were removed from the manuscript

2) Lines 216-228: The experiments with A549 and BMDM cells should be performed also using treatment with itraconazole as a reference antifungal.

Answer: We thank the reviewer for the suggestion. However, we decided to use terbinafine and not itraconazole as a control because, like miltefosine, terbinafine has a fungicidal activity against *Sporothrix*, while itraconazole is a fungistatic drug (Marimon et al., 2008; Borba-Santos et al., 2015b; Nogueira Brilhante et al., 2016; Orofino-Costa et al., 2022). Also, like mentioned in the text, the strain of *S. brasiliensis* that we used for these experiments is resistant to itraconazole (Ishida et al., 2018). So, in order to assess the miltefosine fungicidal activity against *Sporothrix*, we decided to use a fungicidal drug as control.

3) Lines 256-258: The authors should mention that considering *Sporothrix* species, there is insufficient evidence between in vitro and clinical observations about antifungal activity.

Answer: This was now mentioned in the discussion

4) Lines 290-291: The study Zuo et al 2011 (doi:10.1124/mol.111.072322) showed that miltefosine interacts with cytochrome c oxidase disrupting mitochondrial membrane potential in fungal cells, corroborating the localization of miltefosine observed in the present work. Besides, there are many other effects observed in fungal cells after miltefosine exposure described in the literature (doi: 10.1016/j.mycmed.2023.101436).

Answer: These results were added in the discussion

5) Lines 300-301: Why itraconazole was not included in these experiments?

Answer: Because itraconazole is a fungistatic drug, as already explained in the previous question

Lines 319-320: The authors should mention that the activity of miltefosine in sporotrichosis treatments of cats was already evaluated and did not present good results in one study (doi: 10.22456/1679-9216.83639).

Answer: These results were added in the discussion

Line 321: I suggest including a paragraph with a conclusion about the study.

Answer: This was added in the text

Reviewer #2

I have thoroughly reviewed the manuscript titled "Milteforan, a promising veterinary commercial product against feline sporotrichosis". While the manuscript offers valuable insights into the efficacy of Milteforan, there are several points that require the authors' attention to further strengthen the manuscript and enhance its overall quality.

Answer: We thank the reviewer for the comments and suggestions that have considerably improved the manuscript.

1) To bolster the comprehensiveness of the study, I strongly recommend expanding the strain diversity, particularly for MIC and MFC analyses. Sporotrichosis is not a rare disease and it is completely feasible to increase *Sporothrix* strain numbers. Only three strains of each species is not acceptable. In addition, consider including strains with distinct genetic backgrounds, since it is reported that *S. brasiliensis* has considerable genetic diversity (see references with the following DOI: 10.1016/j.jinf.2023.02.034; 10.1093/mmy/myac096; 10.1016/j.fgb.2023.103845) that may change the results found by the authors.

Answer: We thank the reviewer for the suggestion. Unfortunately, in spite of several attempts, we were not able to get additional *S. brasiliensis* and *S. scheckii* clinical or environmental isolates. Actually, some laboratories in Brazil they have sent us some strains but we were not able to recover them or they were heavily contaminated with bacteria. If the reviewer does not mind, we would prefer to report in this study only the three strains from each species we are currently working.

2) It seems there might be a reference to Brilacidin that needs further clarification or elaboration in the manuscript. Brilacidin is presented in the results, but not mentioned in the abstract nor discussed. If brilacidin also has in vitro activity against *S. brasiliensis*, why the authors did not make more experiments with this drug as they did with miltefosine?

Answer: These results were removed, since the focus of the study is miltefosine and as a recommendation from another reviewer

3) Why AMB Micelial MIC were not determined in Table 1? There appears to be a discrepancy regarding the determination of AMB Micelial MIC, which should be addressed for clarity.

Answer: These results were added to the table

4) Table 2 seems to have some entries marked as ND (not determined), which should be explained or resolved.

Answer: The results were added to the table

5) l. 209: shows, instead of showss.

Answer: This mistake was corrected

6) In lines 208-210, authors say: "Treatment of *S. brasiliensis* yeasts with 2, 4, and 8 µg/mL of MFS showss dose-dependent damage of the cells since the PI signal increased with the drug concentration". However, as depicted in Figure 4b, only 8 µg/mL MFS significantly damaged cells. Please rewrite to make this clear.

Answer: This was corrected

7) Authors used lung cells and macrophages in their experiments. Sporotrichosis rarely affects the lungs. Consider diversifying the cell types used in your study to include skin cells, since sporotrichosis is a subcutaneous disease, and cat cells, especially considering the intended application for treating feline sporotrichosis, as mentioned in the title of the study.

Answer: In humans, the main reported clinical forms of sporotrichosis are the cutaneous or lymphocutaneous forms, however, reports of disseminated and extracutaneous infections have increased over the years among immunocompromised patients, mainly in hyperendemic areas, being pulmonary sporotrichosis one of the most common (Kar Aung et al., 2013; Bonifaz and Tirado-Sanchez et al., 2017; Queiroz-Telles et al., 2019; da Silva Ribeiro Gomes et al., 2023). In cats, sporotrichosis is usually more aggressive, and the most commonly found clinical signs are multiple skin nodules and ulcers, frequently associated with nasal lesions. However, systemic disease is not rare, involving many organs, including the respiratory tract and lungs, probably due to the dissemination from the nasal lesions, for which respiratory signs are frequently seen and are usually associated with a higher rate of therapeutic failure (Schubach et al., 2004; Gremiao et al., 2015; de Souza et al., 2018). This is the justification of why using pulmonary epithelial cells in our experiments.

On the other hand, macrophages are part of the main innate immune cells that help to establish a protective immune response against many pathogenic fungi, including *Sporothrix* (Garcia-Carnero et al 2018). For this reason, there are many reports of the interaction between *Sporothrix* spp. and macrophages, and the response of these immune cells to the fungal pathogenic phase (yeasts) is well known (Garcia-Carnero et al., 2018.2; Garcia-Carnero et al., 2019; Vargas-Macías et al., 2022; Gomez-Gaviria et al., 2023), reason for which we decided to use macrophages.

As recommended by the reviewer, we agree that is important and necessary to use more cell types for the evaluation of the MFS activity against *Sporothrix* yeasts, but we decided to focus only in these two cell types for now, since they represent one of the most frequently affected host tissues in systemic infection and one of the immune cells reported to contribute to the immune response against *Sporothrix* spp.

8) There seems to be an issue in Fig 4a that warrants further clarification regarding the presence of hyphae amidst yeasts.

Answer: In our experience working with the different *Sporothrix* species, which is more than 8 years, we have learned that having a 100% of one morphology in a culture is very difficult and highly dependent on the species and strains that are being grown. However, the optimal culture conditions that renders around 98% of yeast-like cells has been standardized (Martínez-Álvarez et al., 2017), conditions which were used for our cultures. Nevertheless, the presence of a small percentage of mycelia is always present, as seen in figure 4. Since in this experiment we could easily distinguish the morphotype that was presenting the PI signal and we only considered the yeast PI⁺ for quantification, we did not see the need to have a pure sample. In the experiments that only yeast cells were needed, we filtered the cultures to obtain only this morphology. We appreciate the reviewer observation and added this information in the methodology.

9) Authors claim in lines 218-219: "As shown in Figure 5a, ML concentrations of 40µg/mL and lower did not reduce A549 cell viability compared to the control." Figure 5a shows that 2.5 µg/ml milteforan significantly reduced cell viability ($P < 0.01$). The statement in lines 218-219 regarding A549 cell viability should be corrected to reflect the significant reduction observed at 2.5 µg/ml of Milteforan, as shown in Figure 5a.

Answer: This was corrected

10) The comparison between ML and TRB treatments in Fig 5b requires further contextualization, considering the significant difference in concentrations used. ML concentration used is eight times higher than TRB concentration! Also, plasma concentration of miltefosine (see PMID: 31685474) is 16 µg/ml after seven days of therapy (200 mg/day), which is 2.5 times lower than the concentration used in this experiment. The plasma concentration of Miltefosine should be considered in the interpretation of the results.

Answer: We decided to use lower concentrations of TRB because, like we show in Table 1, this drug MIC *in vitro* of the *S. brasiliensis* strain that we used for these experiments is 4 times lower (0.5µg/mL for yeast) than the MIC of MFS or ML (2µg/mL). As previously reported in the paper mentioned by the reviewer and in other report (Jimenez-Anton et al., 2017), where the plasma concentration of MFS in mice after oral administration of 20mg/kg at 24 h, which is the time that we incubated the infected epithelial cells, decreases significantly to a concentration around 30µg/mL. This suggests that the plasma concentrations of miltefosine throughout the treatment should be considered. However, these results were obtained in a mice model, whose conditions are very different from an *in vitro* experiment, reason for which we cannot really compare the drug absorption. Nevertheless, taking under consideration the decrease of the drug concentration *in vivo*, we decided to use the two higher concentrations that did not cause toxicity in the cells, 20 and 40µg/mL, to assure a high enough concentration of the drug to observe its fungicidal activity, which as seen in figure 5b, worked. Of course, we propose to perform

these experiments in an animal model, to confirm the results that we see in this study *in vitro*.

11) Figure 6 has the same issue with ML/TRB concentrations mentioned above. Moreover, the concentration units of ML need to be clarified for consistency between the figure and the text.

Answer: The issue with the ML/TRB concentrations was answered in the previous question. The concentration units of ML were corrected in figure 6

12) Assessment of cytokine production: While the assessment of BMDMs is intriguing, it would be more informative to include similar experiments using cat cells to provide a more comprehensive understanding, especially considering the differences in immunology across species. The immune response of cats against *S. brasiliensis* is different from that observed in humans and mice.

Answer: We thank the reviewer for this observation. We completely agree with the reviewer about it. Unfortunately, we do not have access to cat cells and if the reviewer does not mind we would prefer not to do these experiments at this moment.

13) Similar to earlier points, the discrepancy in TRB/ML concentration in Fig 7 should be addressed for accuracy.

Answer: The concentration units of ML were corrected in figure 7

14) Discussion needs to be highly improved. The discussion section (64 lines) appears to be relatively shorter compared to the introduction (90 lines). It would be beneficial to expand upon the discussion to ensure a thorough exploration of the findings and their implications. Also, authors need to explore the limitations of their work and mention future studies needed to make Milteforan a promising veterinary commercial product against feline sporotrichosis, as stated in the title of the manuscript.

Answer: The discussion section was improved with the recommendations of the reviewer

15) I. 440: Providing information regarding the origin and number of passages of the cell lines would enhance the transparency and reproducibility of the study.

Answer: We have added this information to the text: "The cytotoxicity of ML was determined in A549 human lung cancer cells [ATCC, CCL-185, derived from Rio de Janeiro Cell Bank, Brazil (BCRJ-0033) passage (5-10)] using the XTT reduction assay".

Re: Spectrum00474-24R1 (Milteforan, a promising veterinary commercial product against feline sporotrichosis)

Dear Dr. Gustavo H. Goldman:

Your manuscript has been accepted, and I am forwarding it to the ASM production staff for publication. Your paper will first be checked to make sure all elements meet the technical requirements. ASM staff will contact you if anything needs to be revised before copyediting and production can begin. Otherwise, you will be notified when your proofs are ready to be viewed.

Sincerely,
Alexandre Alanio
Editor
Microbiology Spectrum